# Nanocrystalline Yb:YAG-Doped Silica Glass with Good Transmittance and Significant Spectral Performance Enhancements

**DOI:** 10.3390/nano12081263

**Published:** 2022-04-08

**Authors:** Hehe Dong, Yinggang Chen, Yan Jiao, Qinling Zhou, Yue Cheng, Hui Zhang, Yujie Lu, Shikai Wang, Chunlei Yu, Lili Hu

**Affiliations:** 1Key Laboratory of High-Power Laser Materials, Shanghai Institute of Optics and Fine Mechanics, Chinese Academy of Sciences, Shanghai 201800, China; hehedong@siom.ac.cn (H.D.); chenyg206@163.com (Y.C.); jiaoyan@siom.ac.cn (Y.J.); kerryqling2000@mail.siom.ac.cn (Q.Z.); chengyue@siom.ac.cn (Y.C.); zhanghui5067@outlook.com (H.Z.); 1999lyj@sina.cn (Y.L.); 2Hangzhou Institute for Advanced Study, University of Chinese Academy of Sciences, Hangzhou 310024, China

**Keywords:** Yb:YAG nanocrystals, silica glass, microstructure, spectral properties

## Abstract

In this study, Yb:YAG-nanocrystal-doped silica glass with high transmission and excellent spectral properties was successfully prepared using a modified sol–gel method. The X-ray diffraction (XRD), micro-Raman spectroscopy, electron paramagnetic resonance (EPR), transmission electron microscopy (TEM), and high-resolution TEM (HR-TEM) analyses confirmed that the Yb:YAG nanocrystals, with their low content, homogeneous distribution, and small crystal size, directly crystallized into the silica glass network without annealing treatment. In contrast with conventional microcrystalline glass having large particles (>0.1 μm) and a large particle content, nanocrystalline glass with a homogeneous distribution and sizes of ~22 nm had higher optical transmittance and better spectral properties. Compared with Yb^3+^ doped silica glass without nanocrystals, the Yb:YAG-nanocrystal-doped silica glass had a 28% increase in absorption cross-section at 975 nm and a 172% enhanced emission cross-section at 1030 nm without any changes in the spectral pattern of the Yb^3+^ ions in the silica glass. Meanwhile, the Yb:YAG-doped silica glass with large size and high optical quality was easily prepared. Therefore, the Yb:YAG-nanocrystal-doped silica glass is expected to be a promising near-infrared laser material.

## 1. Introduction

The fast-expanding development of laser technology has allowed lasers to reach high-energy and frequency levels [1,2,3]. Examples include the National Ignition Facility (NIF) laser device in the United States, the Super Short Laser Device (SULF) in China, and the Vulcan device in the United Kingdom [4,5,6]. However, it is important to note that the repetition frequency of these lasers and others of the same class worldwide is less than 1 Hz. This lower repetition frequency limits the advancement of high-energy and high-power laser applications such as laser shock intensification [7], petawatt lasers [8], ultra-fast lasers [9], etc. [10,11] The realization of high-energy lasers with high frequency is an urgent requirement [12].

The key to developing a high-energy laser with high frequency is to develop new laser material. A high-energy laser with high frequency will demand excellent thermal shock resistance and a high-emission cross-section of the laser gain materials [13]. Fujimoto et al., after comparing the thermal shock parameters of different laser materials, revealed that silica glass had higher thermal shock properties than YAG, which made it a suitable gain medium substrate material for high-energy, high-frequency lasers. They achieved a laser energy of 37 J in neodymium-doped silica glass and concluded that a laser energy of 100 J for a single pulse and 10 Hz for a heavy frequency could be achieved through further improvement in the neodymium-doped silica glass performance [14,15]. Based on these results, John H. et al. believe that silica glass has great potential for the development of high-frequency lasers [14]. However, the low-emission cross-section of the rare earth (RE) ions in silica glass are problematic, thus seriously hindering their application in high-energy and high-power lasers. Yb-doped laser materials are well-suited for high-power operations because of their low quantum defects, excited state absorption, and quenching effects [16,17]. In addition, the wide-emission bandwidth, long excited-state lifetime, and strong absorption band of approximately 980 nm make Yb-doped laser materials ideal for developing ultrashort-pulse and high-energy diode-pumped lasers [16,17,18,19]. Although Yb:YAG crystals have a high-emission cross-section and excellent thermodynamic properties, they cannot be used in high-energy lasers because of their limited crystal growth size [15].

Based on the background, we proposed to embed Yb:YAG crystals into silica glass. By controlling the size of the Yb:YAG crystals in silica glass, the excellent spectral properties of the crystals and the high optical transmission and thermodynamic properties of the silica glass could be maintained. The emission efficiency was high because of the preferential deposition of RE ions in the nanocrystal phase. It has been established that the difference in refractive index between the crystalline and glass phases (GPs) causes optical scattering, which affects the optical transmittance of the glass. Based on the Mie scattering theory, a high-transmittance nanocrystalline glass is obtained by satisfying one of the following two conditions: (1) a small crystal size, or (2) a small refractive index difference between the matrix glass and crystal [20,21]. Bragliam et al. showed that optical scattering losses are negligible when the crystal size in nanocrystal glass is maintained below 100 nm [20]. Therefore, Yb:YAG-nanocrystal-doped silica glass is expected to be a new generation high-frequency laser material.

In this study, Yb:YAG-nanocrystal-doped silica glass with high transmittance and excellent spectral properties was prepared using a modified sol–gel method. X-ray diffraction (XRD), micro-Raman spectroscopy, high-resolution transmission electron microscopy (HR-TEM) combined with electron energy spectroscopy (EDS), and inverse fast Fourier transform (FFT) were used to systematically evaluate the microstructure of the Yb:YAG-nanocrystal-doped silica glass and crystalline derivatives. The Yb:YAG nanocrystals with a low-content homogeneous distribution and small crystal sizes were crystallized directly into the silica glass network without post-annealing treatment, thus making this a novel study. In contrast with conventional microcrystalline glass with high scattering losses, the Yb:YAG-nanocrystal-doped silica glass had excellent optical transmittance and superior spectral properties [22]. In comparison with Yb^3+^-doped silica glass without nanocrystals, the Yb:YAG nanocrystal-doped silica had a significant increase in the emission cross-section at 1030 nm and the absorption cross-section at 975 nm, without any changes in the silica glass spectral pattern of the Yb^3+^ ions. Therefore, this nanocrystal glass is expected to be a promising near-infrared laser material for further application in high-frequency laser fields. 

## 2. Materials and Methods

Low-nanocrystal-content doping is considered an effective method for obtaining high optical transmittance in silica-based nanocrystalline glass. It is necessary to reduce the doping content of Al_2_O_3_ and Y_2_O_3_ as these are the main components of YAG crystals. The 0.14Yb_2_O_3_-0.5Y_2_O_3_-0.87Al_2_O_3_-0.1BaO-98.39SiO_2_ (YABS, in mol%) and 0.14Yb_2_O_3_-0.5Y_2_O_3_-0.87Al_2_O_3_-98.49SiO_2_ (YAS, in mol%) glass were prepared using a modified sol–gel method [23] that was able to prepare glass of a large size, high homogeneity, and precise composition control, which facilitated the accurate analysis of the spectral and structural properties. BaO was introduced as a nucleating agent into the YABS glass to induce phase separation and crystallization of the glass [24], while a nucleating agent was not introduced into the YAS as a comparison sample. The sample composition test values are listed in Table 1 and the detailed preparation procedure is illustrated in Figure 1. The YbCl_3_·6H_2_O (Sigma-Aldrich, St. Louis, MO, USA, 99.998%), AlCl_3_·6H_2_O (Sigma-Aldrich, St. Louis, MO, USA, 99%), YCl_3_·6H_2_O (Sigma-Aldrich, St. Louis, MO, USA, 99.99%), and BaCl_2_·2H_2_O (Sigma-Aldrich, St. Louis, MO, USA, 99.99%) were dissolved in a mixture of tetraethyl orthosilicate (TEOS, Kermel, Tianjin, China) and C_2_H_5_OH (Kermel, Tianjin, China) at room temperature, sufficiently stirred to obtain transparent silica sols after hydrolysis and polymerization reactions. Thereafter, the homogeneous gels were obtained by heating to 80 °C for 10 h. The gel was heat-treated at 200 °C in an oxygen environment for 20 h to obtain white dry-gel particles. Thereafter, these particles were granulated by ball mill to obtain a white powder, which was sintered at 1720 °C in a vacuum for 1 h, quenched, and rapidly cooled to obtain a transparent bulk glass that could be satisfied with various tests after proper treatment.

A Thermo iCAP 6300 inductively coupled plasma atomic emission spectrometer (ICP-AES, Thermo Fisher, Waltham, MA, USA) was used to determine the contents of Yb, Al, Y, and Ba in the samples. The refractive index of the samples was tested by a 2010/M waveguide prism coupler (Metricon, Raleigh, NC, USA). Based on the Archimedes drainage method, the bulk glass density test was carried out. Raman spectra were tested by a Horiba LabRAM HR Evolution-type Raman spectrometer with a wavelength range of 200–1500 cm^−1^ and an excitation wavelength of 633 nm. An X ‘Pert PRO (Holland Panalytical, Almelo, Netherlands) diffractometer for XRD was used to analyze the glass. A HR-TEM (Tecnai G2, FEI, Tokyo, Japan) equipped with an EDS was used to measure the morphology and size distribution of the nanocrystals in the samples. The HR-TEM imaging detector was targeted at backscattered electrons, and the counting time resulting from the EDS was approximately 180 s. The two-dimensional hyperfine sublevel correlation electron paramagnetic resonance (EPR) spectroscopy (2D-HYSCORE, Kalka, Germany) image of each sample was tested using an E580 pulse EPR detector produced by Bruker with a test temperature of 4 K. The absorption spectrum was tested by a Perkin Elmer Lambda 950 spectrophotometer (Norwalk, CT, USA). The static emission spectra of the fluorescence lifetime were measured by an Edinburgh FLS920 Fluorescence spectrophotometer (Livingston, Scotland, UK). All measurements were performed at room temperature.

## 3. Results and Discussion

### 3.1. Homogeneity of Ion Doping

The homogeneous doping of Y, Al, and RE ions was considered as a prerequisite to achieve a homogeneous distribution of nanocrystals in silica glass. However, the homogeneous doping of co-doped ions in silica glass has been a major challenge [25]. It is difficult for co-doping ions to diffuse and move, even when the silica glass is in the molten state, because of the high melting temperature and high viscosity characteristics of silica glass. In addition, mechanical stirring at high temperatures is challenging. Therefore, Yb^3+^ doped silica glass usually experiences poor homogeneity and spectral properties derived from its poor homogeneity.

In this study, silica glass of a large size (Figure 2a), high homogeneity (Figure 2b), and precise composition control (Table 1) were prepared by a modified sol–gel method [23]. The transmittance curves of the YABS and YAS glass are plotted in Figure 2c; both curves showed a transmittance greater than 85% in the visible and near-infrared regions.

High doping homogeneity provides favorable conditions for the homogeneous distribution of nanocrystals, which improves the optical transmittance of nanocrystalline glass. Figure 2d shows the infrared transmission spectrum of YABS glass, and the inset shows a partial magnification of the absorption band near 3650 cm^−1^. The absorption band in the infrared spectrum near 3650 cm^−1^ was attributed to the hydroxyl group in the glass, which was considered a major challenge in the preparation of silica glass by the conventional sol–gel methods. However, the YABS glass prepared by the modified sol–gel method showed low fluctuations near 3650 cm^−1^, which indicated that hydroxyl groups were effectively removed by the new method and that the low hydroxyl contents facilitated the improvement in the glass spectral properties.

### 3.2. Microstructure Characterization

The XRD patterns and Raman spectra of the YABS and YAS glass are plotted in Figure 3a,b to evaluate whether the YABS glass, with the introduced nucleating agent, contained nanocrystalline structures. The XRD patterns of the YABS and YAS glass were almost identical and both showed amorphous glass structures. However, the Raman spectra of the YABS and YAS glass differed significantly, which indicated that the microscopic local structures of YAS and YABS glass varied significantly. A detailed list of structural vibrations in the Raman spectra of the silica glass and YAG crystals is found in Table 2. 

As shown in Figure 3b, the Raman spectra of the YAS glass were considered a typical glass vibrational structure, while the Raman spectra of the YABS glass were considered a vibrational structure that might contain YAG nanocrystals. The Raman vibration peaks of the YABS glass at 217, 393, and 792 cm^−1^ were considered to correlate with each other, and were very similar to the Raman vibrational peaks of YAG crystals reported in prior literature [22,26,27,28,29,30]. In general, the vibrations at 483 and 603 cm^−1^ were attributed to structural defects in the planar quaternary and ternary rings in the silica glass [16,17]. The vibrations of the YABS glass at 483 and 603 cm^−1^ were significantly weaker than those of the YAS glass, indicating that new network connections had been generated and breakpoints formed in the YABS glass. This suppressed the structural defects of the planar quadrilateral and ternary rings in the silica glass. In addition, the vibration of the Raman spectrum 792 cm^−1^ was due to the overlap of the Y-O stretching vibration and the stretching mode of the Al-O terminal bond in the AlO_4_ species [28,31].

The distribution of REs in the host directly determines the separation of REs and thus their photoluminescence (PL) properties. However, controversy remains regarding the spatial distribution of REs, especially in RE-doped nanocrystalline glass where the cation site size or electrical charge does not match that of the RE dopants. Three different views on the spatial distribution of REs in nanocrystalline glass have been proposed: (1) REs are hosted by the GP rather than by nanocrystals [32,33,34,35,36], (2) REs are distributed at the interface between the nanocrystals and surrounding GP [37,38], and (3) REs are at least partially inside the nanocrystals [39,40,41,42]. The main arguments supporting the first two views are the large size mismatch (40–60%) between the RE ions and the substituted cations, and the absence of additional Stark components in the emission bands, based on crystal field effects [43]. However, studies with simply spectral properties cannot provide convincing evidence for the definitive localization of REs in nanocrystal glass, and may even lead to contradictory conclusions. Therefore, direct methods such as advanced high-resolution transmission electron microscopy equipped with an energy spectrometer are required.

To further investigate the crystalline phase in YABS glass, TEM was used to characterize the microscopic morphology of YABS glass. A homogeneous distribution of nanocrystals in the YABS glass was observed in the YABS glass TEM image as shown in Figure 4a; Figure 4a inset shows the particle size statistics of nanocrystals, which range from 10 to 35 nm with a median value of ~22 nm. The transmission electron dark-field imaging of the YABS sample is shown in Figure 4b, with obvious randomly distributed second-phase particles observed in the silica glass substrate. The HR-TEM image of the white-framed region in Figure 4b is shown in Figure 4c, revealing clearly observable two-dimensional lattice stripes. The FFT image of the white-framed region in Figure 4c is plotted in Figure 4d, showing a clear set of crystal diffraction patterns arranged in amorphous diffuse spots, thus indicating the good crystallinity in its second phase. However, some of the atoms were arranged in a disordered state due to the strong interference from the amorphous field. To further analyze the nanocrystal types, the filtered lattice fringe image of the crystal diffraction pattern IFFT image is shown in Figure 4e. The widths of the six stripes in different directions are 1.59, 1.46, and 1.42 nm, and thus the corresponding crystal plane spacings are 0.266, 0.244, and 0.237 nm, respectively. The three groups of crystal plane spacings, measured in different directions, were 0.266, 0.244, and 0.237 nm, respectively. The three crystal plane sets, combined with the diffraction pattern angle information in Figure 4d, were analyzed and found to correspond to the (420), (134), and ($\bar{3}$41) crystal planes of the YAG (JCPDF: 33-0040) crystals, respectively [44]. In comparison with published microcrystalline glass, those in this study had a low content of nanocrystals of small size, which resulted in few structural variations and interference from the amorphous field of the glass; thus, effective diffraction peaks were not observed in the XRD pattern.

To further determine the chemical composition of the nanocrystals, the region containing the separate nanocrystals in Figure 4b was selected for EDS line scanning; the results are plotted in Figure 5b. The greatest challenge for the line-scan technique is to find a sufficiently thin region where there is little or no overlap between the crystals. A suitable region for line-scanning is shown in Figure 5a, where an isolated crystal is encased in silica glass. A schematic of the ideal distribution curves for bulk and surface doping of REs is shown in the inset of Figure 5a. Figure 5b shows the line-scan distribution curves for Yb, Al, Y, and Si elements. The RE aggregation obviously occurs in the interior of the nanocrystals rather than at the nanocrystals–GP interface and/or in the GP only. It is interesting to note that the results of the line-scan showed the presence of Si atoms inside the nanocrystals. There are two explanations for this phenomenon: (1) nanocrystals are embedded in the GP and a thin GP layer above it, or (2) Si atoms erode the nanocrystal and cause a distortion of the nanocrystal [43]. The results of the line-scans of the crystalline regions concurred with the results of the lattice streak images in Figure 4e. Therefore, it was demonstrated that the nanocrystals permeating the silica substrate in the YABS glass were Yb:YAG nanocrystals. In addition, the percentage occupied by nanocrystals in the YABS glass was estimated to be approximately 0.66% based on the doped Al, Y, and Yb contents combined with the data presented by the line-scan technique for each element [45]. Therefore, we believe that most of the Yb^3+^ ions were confined in the GP of the YABS glass.

Ultra-low temperature EPR has been widely used to study the electron Zeeman splitting in the external magnetic field of the electron magnetic moment and the resonance jumps between energy levels caused by their interaction with the electromagnetic field [46]. The 2D-HYSCORE spectrum can accurately describe the average coordination environment of Yb^3+^ because of the hyperfine coupling between the unpaired electrons of Yb^3+^ and the nonzero nuclear spins of the surrounding nuclei. Figure 6 shows the 2D-HYSCORE spectra of the YABS glass; the Si and Al atom signals were detected in the next-nearest neighbors of Yb^3+^, while Y atoms were not detected because of their non-paramagnetic character. The presence of an Si signal indicated that a portion of Yb^3+^ ions were in the GP, while the presence of an Al signal indicated that a portion of Yb^3+^ ions may have been in the GP or in the nanocrystals. The information obtained by EPR (4K, 2D-HYSCORE) coincided with the results of the XRD, Raman spectroscopy, and TEM (HR-TEM, EDS); the nanocrystals were homogeneously dispersed in the silica glass with most of the REs retained in the GP, although a small fraction entered the interior of the nanocrystals. The small amount of REs entering the interior of nanocrystals do not cause additional crystal field effects on Yb^3+^, and the spectral pattern of Yb^3+^ remained in the GP. However, the spectral properties of Yb^3+^ were greatly enhanced by the presence of Yb:YAG, as discussed in detail.

### 3.3. Spectral Characteristics

The absorption and emission spectra of the YAS and YABS glass are plotted in Figure 7. It has been established that the crystal field effect results in the splitting of the absorption and emission peaks if the REs are located inside the crystal. There was no change in the line shape of the absorption and emission peaks in Figure 7, thus confirming that the RE ions were mostly inside the GP [38]. Through the detailed characterization of the glass structure in this study, we confirmed that the YABS glass contained a small amount of Yb:YAG nanocrystals, while the YAS glass that was used as a comparison glass was a pure GP. Figure 7 clearly shows that the absorption and emission spectra of the YAS glass are significantly lower than YABS glass, which was mainly attributed to the small amount of Yb:YAG nanocrystals contained in the YABS glass as opposed to the YAS glass, which did not contain similar nanocrystals. Moreover, the small amount of Yb:YAG present in the YABS glass was insufficient to prevent most of the Yb^3+^ ions from being in the GP [38,43]. Therefore, YABS glass was significantly better than YAS glass in spectral intensity (absorption and emission spectra), while the spectral pattern of the Yb^3+^ ions in the GP was not significantly affected. A similar phenomenon was previously reported [35,43].

The absorption and emission cross-sections of YABS and YAS glasses are calculated based on the following equations [47]:(1)$$\sigma_{abs}=\frac{2.303*OD}{N_{0}L}$$
(2)$$\sigma_{\mathrm{emi}}=\frac{\lambda^{4}A}{8\pi cn^{2}}*\frac{\lambda I(\lambda )}{\int \lambda I(\lambda )d\lambda }$$
where *OD* is the optical density acquired from the absorption spectrum, *N*_0_ is the Yb^3+^ concentration (ions/cm^3^) depending on the ICP results and glass density, *L* represents the sample thickness, *λ* is the wavelength, n is the refractive index, *c* is the speed of light in a vacuum, *A* is the probability of spontaneous radiation, and *I(λ)* represents the intensity of the corresponding fluorescence spectrum at wavelength *λ*.

It has been established that the larger the emission cross-section of the laser material, the lower the laser threshold and the higher the gain. Figure 8a shows that the overall absorption and emission cross-sections of the YABS glass are higher than those of the YAS glass, which was consistent with the spectral properties displayed by the YABS and YAS glass. The absorption and emission cross-sections of the YABS glass at 975 and 1030 nm of ~2.47 and ~1.44 pm^2^, respectively, were much larger than those of YAS glass at 1.93 and 0.53 pm^2^, respectively. Compared to the YAS glass, the YABS glass had a 28% increase in absorption cross-section at 975 nm and a 172% increase in emission cross-section at 1030 nm. The detailed absorption and emission cross-section values for the YABS glass are shown in Table 3. Moreover, a brief comparison of the absorption and emission cross-sections of Yb^3+^ in different hosts was done with the detailed data presented in Table 4. Although the Yb^3+^ doping content and instrumental errors varied in different reports, a general trend was reflected: the absorption cross-section of YABS glass was higher than that of Yb^3+^-doped silica glass, while the values of the emission cross-section of the YABS glass were between those of the Yb^3+^-doped silica glass and Yb:YAG transparent crystals.

Figure 8b shows the fluorescence lifetime decay curves of Yb^3+^ ions in YABS and YAS at the ^2^F_5/2_ → ^2^F_7/2_ transition. The curves of the YAS samples showed exponential decay, while the curves of the YABS samples showed non-exponential decay. The detailed fluorescence lifetime values are shown in Table 3. The results indicated that only one decay channel existed for Yb^3+^ ions in the YAS glass, while two decay channels existed for the Yb^3+^ ions in the YABS glass, corresponding to the fast (30 μs) and slow (872 μs) decay lifetimes, respectively. The variability of the fluorescence lifetime was related to the microlocal environment, in which the Yb^3+^ ions were located [52]. According to Ramachari et al., this was mainly due to the small distance between the RE ions in the sample and the enhanced interactions [53,54]. Based on the fact that the ionic radii of Y^3+^ and Yb^3+^ are very similar, some Yb^3+^ ions enter the crystalline phase and replace the Y^3+^ ions [52]. The Yb^3+^ ions in the crystalline phase are more closely spaced than in the GP, which is the main reason for the fast decay lifetime of YABS glasses. In addition, it was possible that the doping of the Ba^2+^ ions resulted in a distribution of Yb^3+^ ions partly around the YAG and partly around the Al-O and Si-O. It was observed that the fluorescence lifetime of the Yb^3+^ ions in the YABS samples was lower than that of the YAS samples, which may have been due to a portion of the Yb^3+^ ions entering the YAG crystal and the crystal coordination field affecting the fluorescence lifetime [53,55].

## 4. Conclusions

In this study, Yb:YAG-nanocrystal-doped silica glass with high transmittance and excellent spectral properties were prepared using a modified sol–gel method. The microstructure of nanocrystalline glass has been characterized in detail through XRD and Raman spectroscopy combined with HR-TEM. It was confirmed that low content, small size, and the homogeneous distribution of Yb:YAG nanocrystals crystallize directly into the silica glass network without annealing treatment. This ensured the high transmittance and excellent spectral properties of the nanocrystalline glass. Line-scan technology combined with EPR confirmed that most of the RE ions were in the GP and a small fraction were within the nanocrystals. Although the enriched Yb^3+^ ions were present in the vicinity of the nanocrystals, the content in the nanocrystals was low. The spectral pattern of the Yb^3+^ ions was still dominated by the GP; however, compared with the Yb^3+^-doped silica glass without nanocrystals, the emission cross-section at 1030 nm increased by 172% and the absorption cross-section at 975 nm increased by 28%. Therefore, this nanocrystal glass is expected to be a promising near-infrared laser material for further applications in high-frequency laser fields.

## Figures and Tables

**Figure 1 nanomaterials-12-01263-f001:**
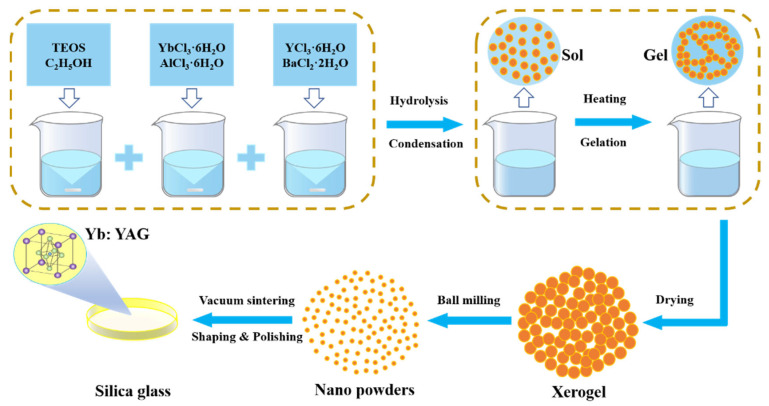
Flow chart for bulk glass preparation using a modified sol–gel method.

**Figure 2 nanomaterials-12-01263-f002:**
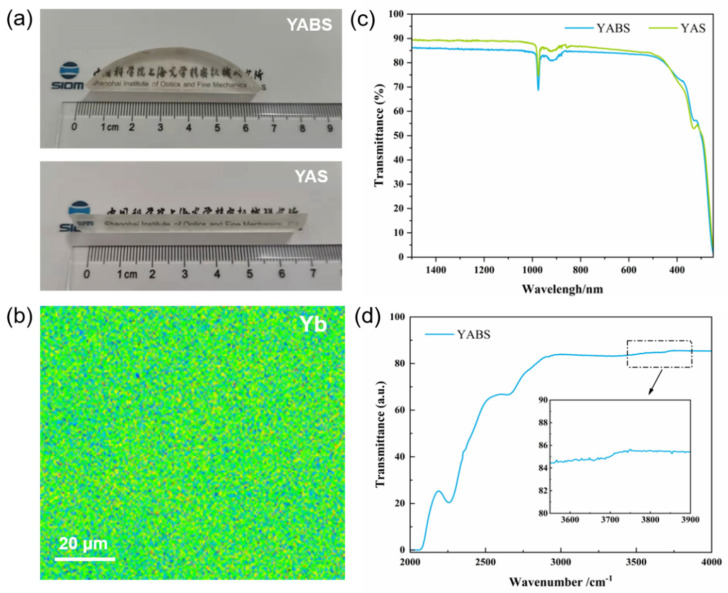
(**a**) YABS and YAS bulk glass after proper processing; (**b**) EPMA surface scan of Yb elements in YABS glasses with a scan window size of about: 200 × 200 μm; (**c**) transmittance curves of YABS and YAS glass; (**d**) infrared transmission spectra of YABS glass, inset shows a local magnification of the absorption band near 3650 cm^-1^.

**Figure 3 nanomaterials-12-01263-f003:**
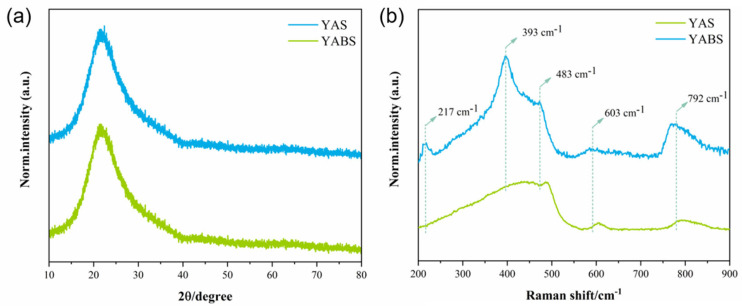
(**a**) XRD patterns and (**b**) Raman spectra of YABS and YAS glass.

**Figure 4 nanomaterials-12-01263-f004:**
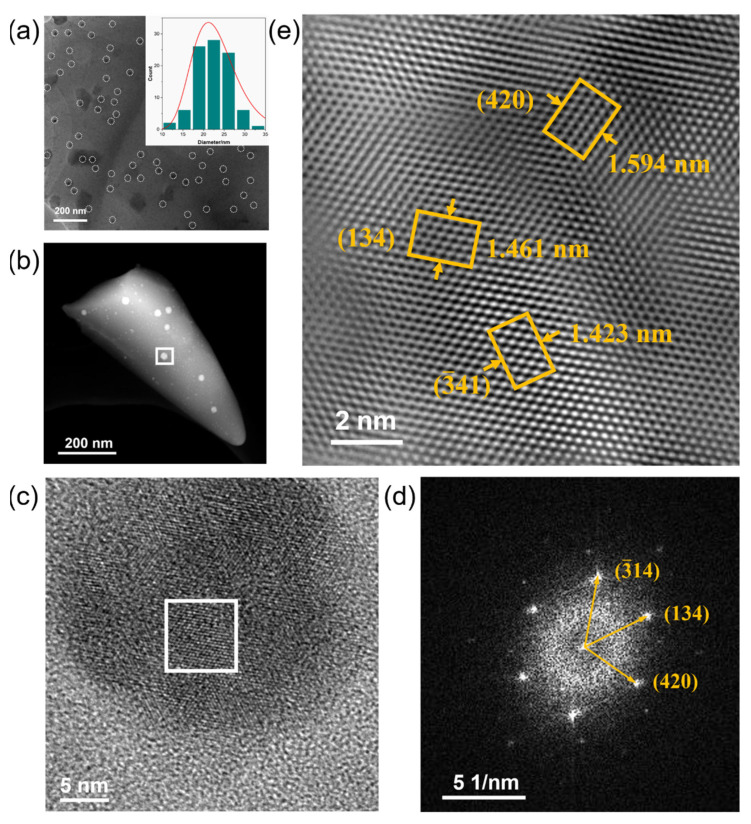
(**a**) TEM image of YABS glass, single nanocrystals are marked with white ellipses, the inset shows the particle size statistics of nanocrystals; (**b**) Transmission electron dark-field imaging of YABS glass; (**c**) Local enlargement of the white frame area in (**b**); (**d**) Fast Fourier Transform (FFT) of the white frame area in (**c**); (**e**) Lattice fringe image filtered by inverse Fourier transform (IFFT) of crystal diffraction pattern.

**Figure 5 nanomaterials-12-01263-f005:**
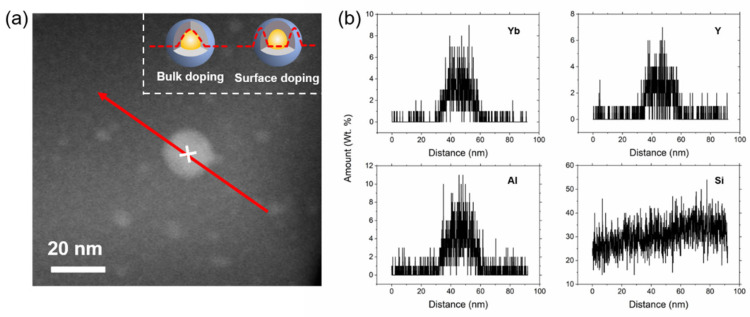
(**a**) Illustration of a typical line-scan for a single nanocrystal. The red line shows the cross-section used for the EDX chemical analysis. A point 50 nm from the beginning of the line is marked with a white cross. The inset shows the ideal distribution curves of bulk and surface doping of REs in nanocrystals. (**b**) Linear profile of the chemical composition, i.e., the amount of Yb, Al, Y, and Si in wt.%.

**Figure 6 nanomaterials-12-01263-f006:**
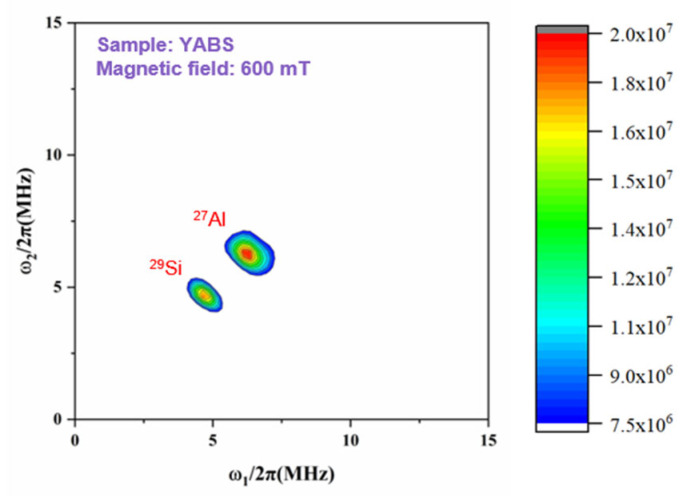
2D-HYSCORE spectra recorded at a magnetic field of 600 mT for YABS glass.

**Figure 7 nanomaterials-12-01263-f007:**
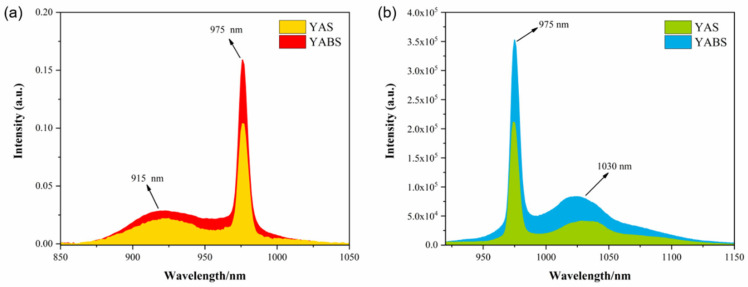
(**a**) Absorption and (**b**) Emission spectra of Yb^3+^ ions in the YAS and YABS glass (λ_ex_ = 896 nm).

**Figure 8 nanomaterials-12-01263-f008:**
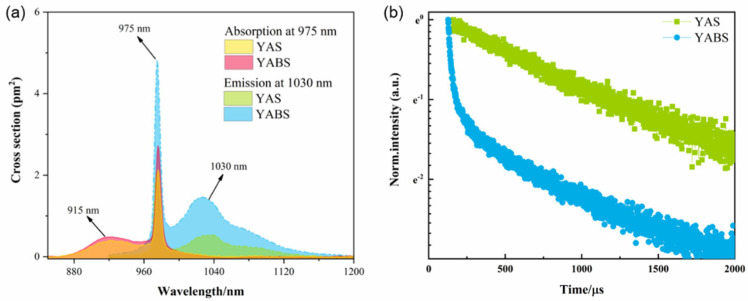
(**a**) Absorption and emission cross-sections and (**b**) Fluorescence lifetime decay curves of Yb^3+^ ions in the YAS and YABS glass (λ_ex_ = 980 nm, λ_em_ = 1030 nm).

**Table 1 nanomaterials-12-01263-t001:** Component test values and physical properties of YABS and YAS glass.

	YABS	YAS
Tested value (mol%)	Yb_2_O_3_/0.14	Yb_2_O_3_/0.15
	Y_2_O_3_/0.45	Y_2_O_3_/0.48
	Al_2_O_3_/0.88	Al_2_O_3_/0.86
	BaO/0.06	-
	SiO_2_/98.47	SiO_2_/98.51
Density (g/cm^3^)	2.2404	2.2238
Refractive index (n)	1.4553 (1064 nm)	1.4532 (1064 nm)

**Table 2 nanomaterials-12-01263-t002:** Structural vibrations in Raman spectra of silica glass and YAG crystals.

Frequency (cm^−1^)	Matrix	Vibrations	Refs.
~217	YAG	Y or Yb translatory	[30]
	Silica glass	-	--
~393	YAG	ν(AlO4)	[27,28,31]
	Silica glass	-	--
~483	YAG	-	--
	Silica glass	Planar quaternary rings structural defect	[16,17]
~603	YAG	δ(Al-O-Al)	[28]
	Silica glass	Planar ternary rings structural defect	[16,17]
~783~800	YAG	ν(AlO4) + δ(Y-O)	[28,31]
	Silica glass	ν(Si-O-Si)	[16,17]

**Table 3 nanomaterials-12-01263-t003:** Absorption cross-section at 975 nm, emission cross-section at 1030 nm, and fluorescence lifetime of Yb^3+^ ions at ^2^F_5/2_ → ^2^F_7/2_ transition for YABS and YAS glass.

	YABS	YAS
Absorption cross-section (pm^2^)	2.47@975 nm	1.93@975 nm
Emission cross-section (pm^2^)	1.44@1030 nm	0.53@1030 nm
Lifetimes (μs)	τ_1_ = 872	τ = 900
	τ_2_ = 30	

**Table 4 nanomaterials-12-01263-t004:** Comparison of calculated absorption cross-sections at 975 nm and emission cross-sections at 1030 nm for Yb^3+^ ions in different hosts with reported values.

	Absorption Cross-Section	Emission Cross-Section	Refs.
	~975 nm (pm^2^)	~1030 nm (pm^2^)	
Yb:YAG single crystal	~0.45	~1.63	[48,49]
Yb:YAG transparent ceramic	~0.78	~1.90	[50,51]
Yb doped silica glass	~2.15	~0.62	[17,46]
Yb:YAG-doped silica glass (YABS)	~2.47	~1.44	In this work

## Data Availability

The data underlying the results presented in this paper are not publicly available at this time but may be obtained from the authors upon reasonable request.

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
