# Peer review of "Nanocrystalline Yb:YAG-Doped Silica Glass with Good Transmittance and Significant Spectral Performance Enhancements"

_nanomaterials, 2022, doi:10.3390/nano12081263_

Round 1

Reviewer 1 Report

The manuscript "Nanocrystalline Yb:YAG doped silica glass with good transmittance and significant spectral performance enhancements " presents interesting experiment results of fabricated glasses. The presented data and discussion are interesting. However, some issues should be explained:

  1. What is the scale of EPMA surface scans presented in Fig. 2, what is the dopants concentrations?
  2. "The XRD patterns of the YABS and YAS glass were almost identical and both showed amorphous glass structures" Please show the evidence of the crystal structure.
  3. What is the distribution of the large crystal presented in Fig. 4a?
  4. What are the excitation spectra of doped YABS and YAS glass?
  5. What is the double/single decay curve difference of YABS and YAS glasses?
  6. What is the application field of presented material in optical fibre technology? How to apply it in the optical fibre preform (it is well known that in such a temperature range the crystalline structure of nanoparticles can be damaged)?
  7. The sol-gel methods often cause some impurities in the final host glass. What was the quality of fabricated glasses?

Reviewer 2 Report

This is a generally well written paper that deserves publication after some minor comments are addressed.

It is a bit surprising that vacuum sintering for 1h did not lead to the occurrence of some crystallization of a silica-rich phase. Please comment.

Why was BaO used as a nucleating agent, instead of an oxide of a high valence cation like Ti4+, Zr4+ or P5+?

Based on the very similar ionic radii of Y3+ and Yb3+, the preferential replacement of Y3+ by Yb3+ in the YAG phase was to be expected (see e.g. J. Ferreira et al., JSST 83 (2017) 436). Please comment.

The fast fluorescence decay of Yb3+ ions observed in the YABS materials with a value of ~ 30 micro-seconds could be due to the fact that the Yb3+ ions which replace Y3+ in the YAG phase become probably much more closely spaced than the Yb3+ ions in the GP and this clustering effect is expected to significantly reduce the emission lifetime. Please discuss.

Reviewer 3 Report

The paper with title: “Nanocrystalline Yb:YAG doped silica glass with good transmit- 2tance and significant spectral performance enhancements “ by Hehe Dong, Yinggang Chen, Jiao Yan, Qinling Zhou, Yue Cheng, Hui Zhang, Yujie Lu, Shikai Wang, Chunlei Yu and Lili Hu presents a beautiful current research focused on study perspective new generation hight-frequency laser materials. In the article, the authors managed to prepare and fully characterize the properties of a new functional material, i.e. silicate glass containing Yb: YAG nanocrystals with size 22 nm.

The scope of article is considerable. Clarity and ordering of the issue are very good. The presented article provides a very good overview of current knowledge about combination of silicate matrix and RE:YAG materials. I mean that article also reflect the focus of the journal. The manuscript is very well written in clear English, the number of citations is sufficient.

Finally, I think there is a great deal of honest experimental work behind these results, and I congratulate the authors on these results. I recommend the manuscript for publication after minor revision mentioned below.

Recommended minor revision:

line 76-92: In that paragraph, I recommend only a description of the preparation or analysis that is used in the article. In my opinion the description of the results is premature here and I recommend to omit it.  

line 130-131: Table 1 – To the value of the refractive index, I recommend add the wavelength at which the value was determined.

line 158: Figure 2(b) - Which colour belongs directly to ytterbium?

line 179: “quadrilat- eral” change to quadrila-teral

line 191:  GP acronym is missing an explanation

line 216: “respec- tively” change to respec- tively.

line 238: “inter-esting” change to intere-sting

line 295: It is not clear which relationship complies with these two laws. I also recommend not mentioning Figure 8 here - it seems confusing.

line 345: Conclusion

It is not entirely clear from the whole article where Yb3+ is located, whether in the glass phase or in nanoparticles. Figure 5 presents the result of the analysis just of one nanoparticle. I interesting for me if more than one of these analyses has been done with the same result? If so, I would recommend including them in the text of article. If the authors are more inclined to the fact that more Yb3+ is present in the glass phase, I wonder what mechanism could then explain the better luminescence properties of the nanocrystalline material? If you have such a hypothesis, it would be nice to include it in the article.

Round 2

Reviewer 1 Report

The manuscript can be accepted in the present form.